# Does Tennis Training Improve Attention? New Approach

**DOI:** 10.3390/children10040728

**Published:** 2023-04-14

**Authors:** Şaban Ünver, İzzet İslamoğlu, Tülin Atan, Metin Yılmaz, Hayati Arslan, Abdurrahim Kaplan, Emre Şimşek

**Affiliations:** 1Faculty of Sports Science Yaşar Doğu, Ondokuz Mayıs University, 55200 Samsun, Turkeyizzet.islamoglu@omu.edu.tr (İ.İ.); takman@omu.edu.tr (T.A.); 2Faculty of Sports Science, Fırat University, 23200 Elazığ, Turkey; 3Faculty of Sports Science, Erciyes University, 38200 Kayseri, Turkey; emresimsek@erciyes.edu.tr; 4Faculty of Sports Science, Hitit University, 19200 Çorum, Turkey; abdurrahimkaplan@hitit.edu.tr

**Keywords:** tennis, attention, training, children

## Abstract

This study aimed to examine the effect of a tennis training program on improving attention. Methods: A total of 40 tennis players from a Tennis Club, 20 in the experimental group (EG) and 20 in the control group (CG), participated in the study. The EG athletes received 40 serve balls from the trainer twice a week for nine weeks. The researcher applied the “d2 attention test” to the EG and CG before and after the nine-week period. Results: After comparing the pretest and posttest attention averages of the experimental group, there was a significant difference in the TN, TN-E, and CP mean scores (*p* < 0.001). In the comparison of the pretest and posttest attention averages of the CG, there was no significant difference in the TN, TN-E, and CP mean scores (*p* > 0.05). The comparison of the pretest attention averages of the EG and CG revealed no significant difference in the TN, TN-E, and CP mean scores (*p* > 0.05). The comparison of the posttest attention averages of the EG and CG revealed a significant difference in the mean scores of TN, TN-E, and CP (*p* < 0.05). There was a statistically significant difference between the posttest–pretest differences in the TN, TN-E, and CP values of the EG and CG (*p* < 0.05). Conclusions: The study concluded that tennis training aimed at developing attention improved the results in the attention test.

## 1. Introduction

Today, tennis, which is an exciting sport that also evokes admiration, is an Olympic sport that has been accepted as a popular sport in Turkey as well as all over the world. Tennis is a sport characterized by the anaerobic energy system comprising a combination of low- and high-intensity movements, including running at different speeds, accelerations, pauses, turns, and shots, throughout the competition [1]. It is an undeniable fact that attention plays a significant role in the correct and effective use of the abovementioned features and abilities during performance. Higher cognitive function (executive control) can also be associated with success in a sports environment [2,3]. Cognitive control is the ability to control one’s attention, behavior, thoughts, and/or emotions in order to override a strong internal predisposition or external distraction and focus on more adaptive and relevant stimuli. [4].

In this day and age, in which people are pushing the limits of human performance, attention is an issue that researchers review. Additionally, studies indicate that sports have positive effects on attention [5,6].

For athletes, attention and decision-making strategies are essential features necessary to achieve good performance in sports. This situation gains more relevance with sports skills that require the athlete to process a lot of information in a short time. Knowing what to pay attention to, how to shift attention to another area if necessary, and how to concentrate attention are basic skills required for proper performance [7].

Attention is a nervous system function, and it enables one to deal with many stimuli according to the need and purpose of the moment [8]. Dividing attention between more than one focal point during an activity emerges as one of the significant factors affecting performance [9]. While an athlete is demonstrating a skill, in addition to the narrowing and broadening of attention, the direction of the focus of attention is crucial in terms of performance. Without this selectivity of attention, the individual would not be able to display a consistent behavior toward many things in its environment. Individuals with a high level of selective attention especially mind the things they need to pay attention to, whereas individuals with low attention or distracted people pay attention to more of the stimuli around them; most of these stimuli are stimulants that individuals should not be concerned about [10].

Tennis is a sport in which attention, concentration, and focus affect performance. One of the most significant factors in being successful in tennis is the athletes’ quality of attention [11]. It is believed that tennis players who do not have good attention quality will fail in matters related to memory transfer as well as experience a decrease in their technical capacity [12]. Accordingly, a high level of attention will inevitably have positive effects on tennis players and be an essential determinant of their performance. For example, attention is critical in effectively returning the serve ball during the match to the opposing side and converting it into points. Attention occurs when internal and external stimuli bombard the cortex with information, and the person can internalize only a certain amount of these stimuli. Therefore, selectivity is required to process only a certain number of stimuli. It is thought that the ability to direct attention to the appropriate stimuli and maintain attention is a significant factor for success in sports. At this point, the relevance of evaluating the attention and concentration the ability of the athlete comes to the fore [13]. Tennis is an open-skill sport that requires making quick decisions under time pressure as well as adapting to ever-changing task demands using creativity and/or problem-solving skills. In this sense, these cognitive skills in tennis require and improve executive function [14]. The current study hypothesizes that tennis training can increase the level of attention; by increasing the level of attention we can assume that this will lead to increased precision. Tennis players need to focus their attention on many targets (e.g., the opponent’s position and the ball), shift their attention in the fastest way possible, and select and sort out the appropriate stimuli. Thus, one of the most crucial factors that determine performance in tennis is the level of attention [11]. The training drill used in this study, which aims to increase the attention level of tennis players, was designed for this study for the first time. In this drill, the researcher asked participants to return the ball to different colored areas using various shot techniques. The fact that this drill was designed with a focus on paying attention demonstrates the originality of the study. This study aimed to examine the effect of a nine-week tennis training program on improving attention and performance.

## 2. Materials and Methods

The experimental and control groups were formed from 40 randomly selected athletes from the Tennis Club that participated in the study. The EG comprised 9 females and 11 males (age: 11.96 ± 1.66 years; height 151.34 ± 1.89 cm; weight 45.12 ± 3.18 kg); the CG had 8 females and 12 males (age: 12.85 ± 2.32 years; height 152.10 ± 1.73 cm; weight 47.29 ± 2.67 kg). The criteria for inclusion in the study were as follows: being a licensed active tennis player for at least five years, regularly undertaking tennis training at least three or four times a week, and not having had an injury or surgery in the last six months. According to Brickenkamp, gender does not affect the d2 test results, so the athletes in the tennis club were randomly selected and divided into groups regardless of their gender.

### 2.1. The Training Program

CG and EG club training (training tennis at least three/week and each training session 90 min in total), continued in the same way. In addition to club training, the EG trained twice a week for nine weeks (18 training sessions in total). In this additional training, the EG had to return 40 regular serve balls from the coach and send the ball to different colored areas determined by the researcher using various shot techniques (forehand or backhand) when the ball first bounced off the ground on their side of the court. As soon as the service ball thrown by the coach landed on the player’s court, the researcher said a random color and the athlete tried to throw the ball into the identified colored area. The serve balls were from both sides (right and left) in a mixed manner. The researcher’s choice of color was made using a random method. However, although it was random, each color was said at least 13 times, in a mixed way. The coach followed the rules when serving the balls (if serving from the right side, the ball had to enter the left service area; if serving from the left side, the ball must enter the right service area) (Figure 1) Additionally, the participants did not return the balls that did not fall into the service area and went out. The researcher applied the d2 attention test, which was developed by Brickenkamp [15] and adapted into Turkish by Toker [16], to the EG before and after the nine-week training period. The CG only continued their club training and were given the d2 attention test.

### 2.2. The d2 Attention Test

The d2 attention test measures selective attention over time. The test page comprises 14 lines with 47 marked letters each. There are 16 different letters in each line, comprising the letters “*p*” and “d” with one, two, three, or four adjacent marks. During the test, the subject has to scan through the lines to find and cross out any letter “d” that is marked with two signs, ignoring other unrelated letters. The subject is given 20 s for each line. The subject is expected to make both quick and accurate markings within 20 s. (Figure 2) As a result of the test of attention the researcher evaluated the parameters of TN (total number of marked items), TN—E (total item number—total error (TN-(E1 + E2)), and CP (concentration performance, number of correctly marked items).

The internal test–retest reliability of the d2 test has been shown to be very high for all parameters (0.95–0.98) and the criterion, construct and predictive validity have been documented and shown to be strong and stable [17]. In 2004, Bates and Lemay sought to replicate the authors’ original reliability and validity reports because the d2 Test of Attention is such a well-used neuropsychological tool in Europe [18]. They found it to be an internally consistent and valid measure with coefficients nearly identical to those previously reported. The measure was scored using the procedures outlined in the manual. (Brickenkamp and Zillmer, [17] which are described below:

#### 2.2.1. Errors

Errors of omission (EO). The total number of d2 target symbols that were processed, but not crossed out across all 14 trials.

Errors of commission (EC). The total number of distracter symbols characters that were processed and crossed out across all 14 trials.

Total number of errors (TE). The sum of EO and EC.

#### 2.2.2. Selective Attention

Processing speed/Total number of items processed (TN). The last (either correctly or incorrectly) crossed out letter in each row was considered the last item processed in that trial. TN was the sum of the total number of characters processed across all 14 trials.

Total number of correctly processed items (TC). TN-TE.

### 2.3. Statistics

Statistical analysis of the data obtained in the research was carried out using the SPSS 21 package program. The study used the Shapiro–Wilk test and determined that the data showed normal distribution. Accordingly, a paired *t*-test was used in the analysis of the data. Statistical significance was set at α = 0.05 for this study. In addition, the researcher calculated the difference between the posttest and pretest values of the groups and used the independent samples *t*-test to compare the differences between the means. The percentage change rate between the pretest and posttest was calculated.

The percentage change rate was obtained by dividing the difference between the posttest and the pretest by the pretest value and multiplying the result by 100.

## 3. Results

In this study, no significant difference was found between the mean age of the CG and EG.

Figure 3 demonstrates that there was a statistically significant difference in the mean scores of TN, TN-E, and CP between the pretest and posttest attention scores of the EG (*p* < 0.001). After comparing the pretest and posttest attention mean scores of the control group, there was no statistically significant difference in the mean scores of TN, TN-E, and CP (*p* > 0.05). The percentage change in attention between the pre-test and posttest is given in the figure.

Figure 4 demonstrates that there was no statistically significant difference in the mean scores of TN, TN-E, and CP in the comparison of the EG and CG pretest attention scores (*p* > 0.05). It was observed that there was a statistically significant difference in the mean scores of TN, TN-E, and CP in the posttest attention mean scores of the EG and CG (*p* < 0.05).

## 4. Discussion

The current study hypothesizes that tennis training can increase the level of attention and by increasing the level of attention we can assume that this will lead to increased precision. Tennis players need to focus their attention on many targets (e.g., the opponent’s position and the ball), shift their attention in the fastest way possible, and select and sort out the appropriate stimuli. Thus, one of the most crucial factors that determine performance in tennis is the level of attention. The training drill used in this study, which aimed to increase the attention level of tennis players, was designed for this study for the first time. In this drill, the researcher asked participants to return the ball to different colored areas using various shot techniques. The fact that the drill was designed with a focus on paying attention demonstrates the originality of the study. This study aimed to examine the effect of a nine-week tennis training program on improving attention.

Both the EG and CG took the d2 attention test. There was a significant increase in the attention scores between the pretest and posttest (TN, TN-E, and CP) in the EG. The study determined that the pretest and posttest attention scores of the CG did not change. When the change rate was analyzed, the EG’s TN value increased by 12.04%; the TN-E value increased by 13.23%; and the CP value increased by 15.45%. In the CG, the TN value increased by 0.64%; the TN-E value increased by 1.04%; and the CP value increased by 0.76%. This result demonstrates that, in addition to routine tennis training, the nine-week tennis training improves attention. This result indicates that attention training for tennis also improves the general attention characteristics of the athletes outside their sports branch. We can assume that an increase in attention will lead to an increase in precision. Based on this, it can be said that attention-oriented sports activities can help children focus their attention. It is believed that this type of training can be useful in eliminating distractions, especially in people with low attention span.

Studies in the literature demonstrate that people can train their attention ability using various methods in populations with different characteristics [19,20,21]. According to the model proposed by Howie and Pate [22], PA has stated that it affects children’s cognitive functions (attention, memory, intelligence quotient (IQ), etc.). Similarly, it was found that the attention of children improved after the exercise program applied in our study.

There are many studies in the literature that have parallels with the results of this study [23,24,25]. In their study on national tennis players, Keller et al. stated that the external attention focus was higher as a percentage when testing individuals’ serving shots to the target area [23]. These studies found that training or exercise had a positive effect on attention. In their study on tennis players aged 8–9, they reported that the athletes in the external attention focus group had better attention scores in the posttest [24].

They reported that there was a significant improvement in the attention and concentration performance of the participants in their research, in which they applied 10 min of coordinated exercise to the experimental group students aged between 13 and 16 years in addition to routine physical education lessons [26]. In another study that Reigal conducted with male football players, they administered an attention training program in addition to routine soccer training and reported that there was a difference between the pretest and posttest attention (TN, TN-E, and CP) scores of the experimental group that received the training [27]. A different study conducted with football players stated that selective attention and concentration scores improved with similar results [28]. The studies of Drozdowska et al. showed that physical fitness was associated with improved attention and memory functions in children [29].

It has been stated that physical activities and exercise positively affect children’s attention, concentration, reading, and mathematics achievement [30,31,32]. In a study of children with attention deficit disorder, Verret et al. stated that the physical training program improved the muscle capacity, motor skills, and attention of these children [33]. Other reports have stated that attention skills exercises and educational games carried out for a certain amount of time could contribute positively to the attention development of children [34,35]. In their study on children’s attention training, Clikeman et al. [36] found that there was an improvement in the visual and auditory attention of the children in the attention training group.

This study determined that the pretest d2 attention levels of the two groups were similar, but the posttest attention levels of the EG were higher. This result indicates that attention-oriented tennis training can improve the athlete’s attention and the accuracy of their shooting performance in tennis. Because athletes with increased attention levels eliminate different external stimuli more quickly, they can also experience an increase in the accuracy of forehand or backhand shots. We can assume that an increase in attention will lead to an increase in precision.

Studies in the literature have parallels with the findings of this study. In a study on tennis players aged 8–9 years, it was reported that there were significant differences in favor of extrinsic attention-oriented athletes compared to the athletes in the control group [24]. In the study in which an educational game program was applied to children for eight weeks, Akandere et al. [25] found no difference between the pretest attention values of the groups; however, they did find a difference between the groups’ posttest attention values. In his study, in which he applied 12-week life kinetic training to young male basketball players, Vural [37] found that there was no difference in the pretest attention (TN) value of the groups, but that there was a difference in the posttest attention (TN) value of the groups in favor of the experimental group.

Studies examining children who participate and do not participate in different sports branches show that the attention values of children who do sports are higher [5,38,39].

In their study with male football players, they administered an attention training program in addition to the routine football training for the experimental group, and they stated that there was a difference in the posttest attention (TN-E, CP) values of the control and experimental groups, but there was no difference in the TN value. In the current study, however, there was a difference in the TN value. It is believed that the different results in this parameter are due to the training type and the sample group [27].

### Strengths and Limitations

The main strength of this work is its design. Our study is one of the few to examine the relationship between cognition and sports performance using a domain-based approach. Most of the work in this area has been done in a laboratory setting. Therefore, this study extends previous knowledge on the relationships between exercise, cognition, and physical performance in laboratory settings by suggesting practical sports applications.

This study has the following limitations. First, not serving the ball with the ball thrower in the practice. Second, future research with a larger sample size is needed to clarify the relationship between cognitive function and tennis performance. Third, the training time was only 9 weeks, which may have prevented us from seeing more robust results on some measures. More protracted programs would probably offer more significant changes in the measures of attention assessed.

## 5. Conclusions

Tennis is a branch of sports that requires a high level of attention during training and competition. Therefore, tennis players participated in a tennis training program aimed at improving their attention for a higher level of performance. As a result of the nine-week tennis training program implemented within the scope of the research, there was an improvement in attention levels. In addition to the drill, which was designed to improve the attention level in our study, training programs may contribute to developing the attention levels and performances of athletes. With this study, the necessity of doing such attention studies has emerged in branches of sports where attention is important. It is recommended that similar drills be administered to athletes from different age groups and in different sports branches to improve their attention. It is suggested that tennis players could plan different training/exercise activities rather than just attention training for tennis.

## Figures and Tables

**Figure 1 children-10-00728-f001:**
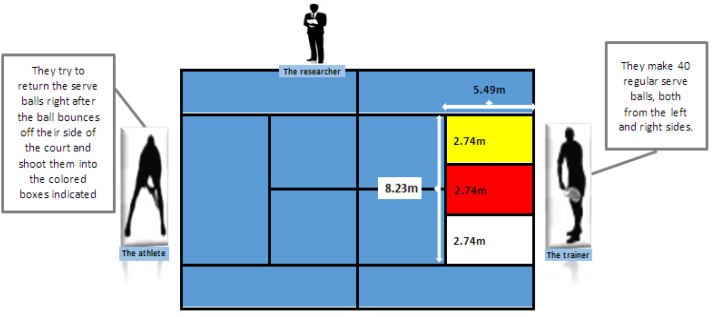
The design of the training.

**Figure 2 children-10-00728-f002:**
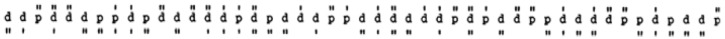
Sample d2 task trial.

**Figure 3 children-10-00728-f003:**
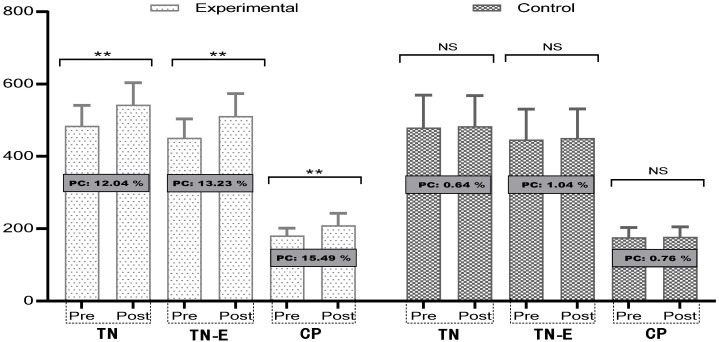
The comparison of the pretest and posttest attention scores within groups and percentage change. ** *p* < 0.001, NS: non-significant, Pre: pre-test, Post: posttest, TN: total number of marked items, TN-E: total item number−total error, CP: concentration performance, number of correctly marked items, PC: Percentage changes pre and posttest.

**Figure 4 children-10-00728-f004:**
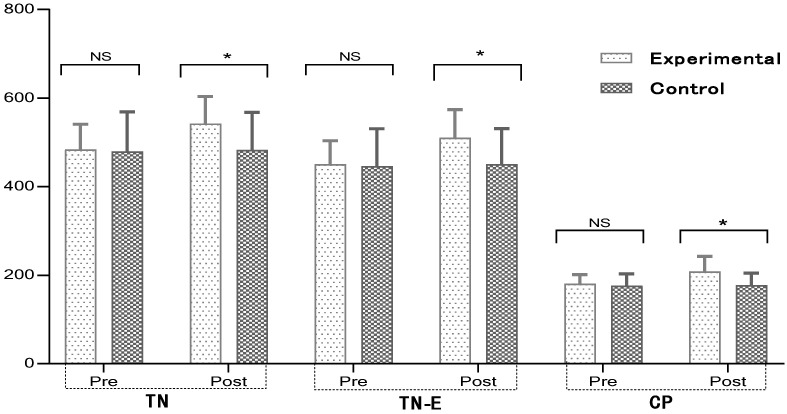
The comparison of the pretest and posttest attention scores between groups. * *p* < 0.05, NS: non-significant, Pre: pre-test, Post: posttest, TN: total number of marked items, TN-E: total item number−total error, CP: concentration performance, number of correctly marked items.

## Data Availability

The data used to support the findings of the current study are available from the corresponding author upon request.

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
