# Peer review of "Does Tennis Training Improve Attention? New Approach"

_children, 2023, doi:10.3390/children10040728_

Round 1

Reviewer 1 Report

This study aimed to examine the effect of a tennis training program on improving attention.  There are a total of 40 tennis players from the Tennis Club, 20 in the experimental group and 20 in the control group, participated in the study. The study concluded that tennis training aimed at developing attention improved the results in the attention test and increased the number of accurate shots. This study has some research contributions for motor learning field in sport science. There are some comments as follows:

1.      This study aimed to examine the effect of a tennis training program on improving attention. There are few literature reviews about the effect of a tennis training program on improving attention. The researchers should demonstrate more past original reports and evidence-based researches in order to deeply review to look for research gap in this study, which could provide more deeply review in the introduction.

2.      Please demonstrate research hypotheses in the introduction.

3.      Page 2. In section of Materials and Methods, please demonstrate some information about IRB(institutional review board) , including how to recruit participants and informed consent process

4.      Page 3. Statistics method use mixed-design ANOVA that is better method in this study. Because based on Page 2, this study showed that “ the experimental and control groups were formed with 40 randomly selected athletes from the Tennis Club who participated in the study.” Therefore, it is better to use the statistics method of mixed-design ANOVA to analysis data.

5.      Page 3, if statistical significance was set at α = 0.05 for this study, please write it in the statistics section.

6.      Page 6, in discussion shows that “Studies in the literature demonstrate that people can train their attention ability with various methods in populations with different characteristics [12-14]. The findings of these studies support this research. Studies indicate that regular physical activity not only contributes to health [15], but also has a beneficial effect on the children's cognitive functions, such as attention [16], and academic performance [17].” Please provide the theory of brain development and cognitive function related sport and exercise to explain why these research hypotheses could be confirmed in this study.

7.      Page 7, in conclusion section, please indicate what research contributions for findings of this study and what are suggestions of further study

8.      Some references of this study are old, please search new empirical reports in 2019-2023 about research topic.

Author Response

Dear Reviewer;

Your suggestions were very important to us to make our work more understandable and valuable.

Thank you for your special interest and comments.

Best regards.

Reviewer 2 Report

Line 55

One of the most significant factors in being successful in tennis is the athletes’ quality of attention.

This statement needs a reference.

Line 64

The ability to direct attention to the appropriate stimuli and maintain attention is a significant factor for success in sports.

This statement needs a reference.

Line 66

The current study hypothesizes that tennis training can increase the level of attention and the number of accurate shots in tennis.

Tennis training increases the number of accurate shots in tennis, is logical and does not need to be emphasized.

I think this is more correct:

The current study hypothesizes that tennis training can increase the level of attention, by increasing the level of attention we can assume that this will lead to increased precision.

Line 70

Thus, one of the most crucial factors that determine performance in tennis is the level of attention.

This statement needs a reference.

Line 79

The experimental group com- 79 prised 9 females, 11 males (age: 11.6 ± 1.66 years); the control group had 8 females, 12 80 males (age: 12.85 ± 2.32 years) tennis player participants.

The years of the experimental group should be written with two decimal places.

Experimental group 11,6 control group 12.85 years. Should it be checked whether the difference in age is statistically significant?

It would be good to better describe the sample: height, weight, training experience...

The Training Program

Add information how long before the serve does the coach determine the field where the tennis player should shoot.

It should be emphasized that the training of the experimental and control groups lasted the same.

The D2 Attention Test

The reliability of the test on the sample of athletes was not measured in this paper, so a couple of references should be added that confirm that the test is reliable on the sample of athletes.

Figure 2 should be deleted. The topic of the paper is does tennis training improve attention, not precision. Another reason is that the control group was not tested for accuracy. The information that tennis players have significantly improved their accuracy can be commented on in the discussion and one sentence in the results is fine. The training program is a drill and not a measuring instrument for checking precision. If it is a measuring instrument, then its metric characteristics should be checked first.

Discussion

Not a single validated test to assess the accuracy of the shot was used, but a drill designed to increase attention. The impact on precision should be discussed cautiously. Therefore, wherever authors mention an increase in precision, they must use the phrase, we can assume that an increase in attention will lead to an increase in precision.

Author Response

(The authors gave the same response as above.)

Round 2

Reviewer 2 Report

The paper has been corrected according to the recommendations, and can be published.